# A Structural Equation Model for Understanding the Relationship between Cognitive Reserve, Autonomy, Depression and Quality of Life in Aging

**DOI:** 10.3390/ijerph21091117

**Published:** 2024-08-23

**Authors:** Maria Gattuso, Stefania Butti, Inaihá Laureano Benincá, Andrea Greco, Michela Di Trani, Francesca Morganti

**Affiliations:** 1Department of Human and Social Sciences, University of Bergamo, 24129 Bergamo, Italy; stefania.butti@unibg.it (S.B.); inaiha.laureanobeninca@unibg.it (I.L.B.); andrea.greco@unibg.it (A.G.); francesca.morganti@unibg.it (F.M.); 2Department of Dynamic and Clinical Psychology, and Health Studies, Sapienza University of Rome, 00185 Rome, Italy; michela.ditrani@uniroma1.it; 3CHL—Centre for Healthy Longevity, University of Bergamo, 24129 Bergamo, Italy

**Keywords:** aging, health, quality of life, depression, cognitive reserve, SEM

## Abstract

In recent years, aging has become a focal point of scientific research and health policies due to the growing demographic trend of an aging worldwide population. Understanding the protective and risk factors that influence aging trajectories is crucial for designing targeted interventions that support healthy aging and improve people’s quality of life. The aim of this study was to explore the relationships between variables of aging. A total of 103 Italian participants (55–75 years old) underwent multidimensional assessments that covered cognitive, functional, emotional, and quality of life dimensions. Structural equation modeling was used to analyze the data and elucidate the relationships between depression, quality of life, cognitive reserve, executive functions, and daily autonomy. The findings revealed that a higher quality of life was associated with reduced depressive symptoms. In addition, cognitive reserve emerged as a protective factor positively correlated with both quality of life and daily autonomy. In this study, quality of life was determined using physical health, psychological, social relationships, and environmental domains. Identifying the significant relationships between these variables in a sample of late adults and young-aged people has given us useful elements for designing psycho-educational interventions that can be aimed at preventing frailty in later old age or supporting healthy longevity.

## 1. Introduction

Aging is a complex and interconnected process through which each individual is required to remodel all specific characteristics and capacities related to the psychological, biological, social, and environmental components that they have built and refined throughout their life.

Currently, this topic has become of great importance in the international scientific landscape. Given the increase in life expectancy in developed countries and demographic projections which indicate the growing presence of old and very old individuals and, conversely, the increased risk of developing frailty and disease at this stage of life [1], the following two questions are important to consider:How can we support individuals who are aging or approaching old age in achieving the best possible health conditions, well-being, and quality of life?How can we raise individuals’ awareness of their aging process and promote their potential and resources to enable them to take full responsibility for their well-being in old age [2,3,4]?

To answer these questions, it is necessary to take a step back. Because of the complexity of this issue, it is deemed extremely important to define as precisely as possible which indicators positively or negatively influence healthy longevity trajectories.

Starting from a healthy aging perspective, the key model of Rowe and Kahn [5] identifies as positive factors a low risk of disease or disability, high physical and functional abilities, high cognitive functioning, and an active attitude. Exploring the mechanisms underlying vulnerability or health capacity in aging is challenging [6], but studies have investigated the relationship between active aging and quality of life. Quality of life is defined by the World Health Organization [7] as the perception that individuals have of their position in life in the context of the culture and value systems in which they live and in relation to their goals, expectations, standards, and concerns. It is a broad concept that is complexly influenced by physical health, psychological state, degree of independence, social relationships, and connections with salient features of the environment.

A systematic review of 26 studies conducted between 2000 and 2020 on individuals aged 60 and older highlighted a positive and consistent association between active aging and quality of life in its various domains [8]. In an attempt to specify the indicators that influence this relationship, a representative study of the Spanish population aged 50 and over found that higher levels of cognitive reserve were associated with a higher quality of life. Furthermore, this relationship was found to be mediated by psychological factors such as depression and cognition [9]. Described by Stern [10] as the brain’s ability to use different neural networks and cognitive strategies, a high level of cognitive reserve could enable older individuals to cope effectively with the changes and adaptations required at this stage of life.

The mediating role of depression has been investigated in other studies, confirming that depressive disorders have a negative impact on quality of life [11] which is characterized by a negative and significant relationship between depression and quality of life scores [12]. Depressive symptoms were investigated together with autonomy levels in daily activities in an older European population who had multiple comorbid conditions, and it was found that limitations in this ability to be autonomous are crucial intermediary factors in the association between multimorbidity and a negative quality of life in people who have two or more chronic diseases [13].

According to Kazama et al. [14], depressive symptoms can predict a future decline in daily autonomy and affective disorders and depression are correlated with cognitive impairment [15]. In addition, lower depressive symptoms together with a higher cognitive reserve are associated with a self-perception of successful aging [16].

Although the association between cognitive impairment and frailty is well established [17], depression seems to play a decisive role in the development of a type of frailty composed of a loss of energy, tiredness, and poor sleep [18]. Fragility and depression appear to be interrelated so much so that, according to the authors in [19], one represents a risk factor for the other [20].

Furthermore, depression was found to also be a mediator in the relationship between cognitive functioning and quality of life [21]. However, this finding was obtained in a study involving breast cancer patients aged between 39 and 60 and over, and thus not on a general aging population. However, these findings contribute to a better understanding of the relationship between depression, cognition, and quality of life. Therefore, depression is partly responsible for the effect that cognitive decline has on an individual’s quality of life. Furthermore, the same study highlighted that the mediation of depression was greater in the relationship between objective cognitive functioning and quality of life compared to that between subjective cognitive functioning and quality of life.

Finally, a study [22] conducted with the aim of explaining the relationship between quality of life and cognitive functions, anxiety, and depression in hospitalized elderly patients has shown that cognitive deficits, depression, and other mental and physical illnesses negatively influence quality of life. More specifically, the authors reported that levels of depression and anxiety were negatively correlated with quality of life.

The type of relationship between sociodemographic factors such as age and gender in depression, the main outcome of our study, has been analyzed by various studies. Based on these, depression diagnoses, as well as the presence of depressive symptoms, vary with age and the gender difference remains notable and consistent up to approximately 55 years of age [23]. However, it remains unclear whether the impact of gender on depression begins to change at the age of 55 years or later (as soon as one reaches the age of 65 years, for example) [24]. Indeed, it is possible to find either an absence or a non-significant presence of a gender difference in older adults with depressive disorders [25], suggesting that the gender difference in depression may decrease with age. Otherwise, a greater number of diagnoses of depression or higher levels of depressive symptoms have been found in women compared to men over the age of 65 [26].

Based on the aforementioned, we note how these factors have consistent effects on each other and can guide us in better understanding how frailty and disease develop during aging. However, there does not appear to be a comprehensive analytical approach that relates all the above-mentioned factors, which we have chosen to include in our model to analyze their contribution in guiding aging trajectories. Therefore, starting from the point of view of health over a person’s life span, we investigate these existing relationships to understand how, in an aging population, they can influence growth paths that in the future could tend towards potential fragility rather than towards healthy longevity.

Given that, the present study aims to investigate the relationships between quality of life; psychological factors, such as depression and functional abilities; and cognitive factors, such as frontal function and cognitive reserve, in the phenomenon of aging. Furthermore, it is noteworthy that many studies have analyzed the impact, relationship, or influence of depression on other factors; for example, quality of life or cognitive decline. Additionally, a relationship between cognitive reserve and quality of life was observed. Conversely to this, with our study we intend to observe the impact of the aforementioned factors on depression, which has been considered to be one of the most considerable risk factors for the development of frailty and unhealthy conditions. Specifically, we hypothesize that the level of perceived quality of life, fluid and crystallized cognitive functioning, and daily autonomy, together with socio-demographic variables such as gender and age, will have a significant impact and correlation with a person’s emotional state.

## 2. Materials and Methods

### 2.1. Participants

A total of 103 citizens of Bergamo, a middle-sized city in northern Italy, were recruited on a voluntary basis from the general population and divided into two groups: one aged 55–65 years (n = 52) and categorized as late adult and the other aged 66–75 (n = 51) and categorized as old adult. In the first group, 80.77% were female and the median age was 59.3 years (IQR = 3.44). In the second group, 78.43% were female and the median age was 70.4 years (IQR = 2.67).

The sample was divided into these two groups to carry out descriptive and comparative analyses, with the aim of analyzing the differences between the two age groups and the trend in the participants’ performance of cognitive tasks, emotional measures, autonomy, and quality of life. Specifically, this division was necessary for the descriptive analysis because some different instruments were used, depending on the age of the participants, to measure the same construct or the same ability.

Participants were recruited based on self-selection within social association centers for the elderly and through the dissemination of the initiative via flyers. The inclusion criteria were being aged between 55 and 75 and being a resident or domiciled in the city of Bergamo. The only exclusion criterion was having severe physical impediments that would have made it impossible to perform some of the cognitive tasks.

After signing informed consent forms, the participants underwent a multidimensional assessment of their cognitive functioning, emotional state, functional level, and perceived quality of life. The batteries and instruments used are described below.

The missing data collected concern the WHOQOL-brief and were handled according to WHO, 2000 [27].

This study was approved by the Ethics Committee of the University of Bergamo, Italy, in accordance with the ethical principles established in the “Declaration of Helsinki” and in the “Convention on Human Rights and Biomedicine”—Oviedo Convention.

### 2.2. Instruments

The Cognitive Reserve Index Questionnaire (CRIq; [28]) estimates an individual’s cognitive reserve by collecting information related to their adult life. The questionnaire consists of three sections. The first section, CRI-School, records the level of education attained by the individual. Here, 1 point is awarded for each year of schooling completed, and 0.5 points are assigned for years of repetition. The second section, CRI-Work, records the type and number of years of employment undertaken by the individual. In the third section, CRI-Leisure Time, the frequency (rarely/never or often/always) and number of years devoted to activities typically carried out outside of work or school hours are collected. Each section provides a final score that contributes to determining a total CRI score of 70 or less (lower) to 130 or more (higher).

The Frontal Assessment Battery (FAB15; [29]) was used to screen general executive functioning. Its five tasks evaluate the ability of the participant to complete similarity, phonemic fluency, motor series, contradictory instructions, and go-no-go tasks—namely, their inhibition ability. The maximum score is 15 (a maximum of 3 points for each task), and the cut-off is set at 9.36/15. The score is adjusted for gender and age.

While the previous rating scales were used for the entire sample, in this study, specifically designed autonomy and depression rating scales were administered to the different age cohorts (55–65 and 66–75 years).

For the 66–75-year-old group, we used the Instrumental Activity Daily Living Scale (IADL; [30]), which investigates old people’s ability to perform functional activities necessary to maintain their independence. The scale consists of eight complex actions (using the phone, going shopping, preparing meals, taking care of the house, doing laundry, using public transportation, taking medication, and managing money). The maximum score that can be obtained is 8 (total independence), and the minimum is 0 (not independent).

For the 55–65-year-old group, we used the Advanced Instrumental Activities of Daily Living Scale (AIADL; [31]), which can be considered an evolution of the AIDL scales, having been updated to include some advanced activities related to the use of technology or the internet. It consists of seven abilities (the use of mobile and landline phones; shopping; the use of a computer; the use of a television and a DVD player; taking care of the home, including the use of household appliances; responsibility and skill when using money and making bank deposits; and driving). For each skill, a six-point Likert response scale is used, ranging from 0 (not capable or not autonomous) to 5 (very autonomous and capable). The maximum score that can be obtained is 35, and the minimum is 0. In addition, for each skill tested, there is a question to assess whether the skill was previously present (yes = 1) or not (no = 0).

For the 66–75-year-old group, we used the Geriatric Depression Scale (GDS; [32]), which records depressive symptoms among elderly people. It consists of 15 items. A score between 0 and 5 is to be considered an absence of depression.

For the 55–65-year-old group, we used the Beck Depression Inventory-II (BDI-II, [33]). It is a self-report scale that assesses the severity of depression in adult and adolescent patients aged 13 years and older. It consists of 21 items. A score between 0 and 13 is considered to indicate no or minimal depression.

The WHOQOL-BREF [27] is a self-administered questionnaire. It consists of 26 questions that assess an individual’s perceived health and well-being in relation to the previous two weeks. Questions are answered on a 1–5 Likert scale. The items cover four domains: the physical health (Dom 1), psychological (Dom 2), social relationships (Dom 3), and environmental (Dom 4) domains. Each of them can be scored from 4 to 20. Ratings are not valid if more than 20% of the data are missing. When the answer to only one item is missing within a domain, it is replaced with the average of the other items belonging to it.

### 2.3. Statistical Analysis

A descriptive analysis was carried out to describe the participants’ cognitive, emotional, and functional abilities and self-perception of their quality of life. Distribution normality was tested using the Shapiro–Wilk test for normality. In addition, the Frontal Assessment Battery’s 15 scores were adjusted according to Ilardi et al. [29]. 

The software R 4.3.3 (R Foundation for Statistical Computing, Vienna, Austria) was used to analyze all data. 

To compare age groups (55–65 and 66–75), we performed t-tests with a Bonferroni correction for multiple comparisons of the following variables: autonomy scores, CRIQ total scores, depression scores, quality of life (including the domains of physical health, psychological, social relationships, and environment), and FAB15 scores (*p* value cutoff = 0.05/8). When the assumption of a normal distribution was violated, Mann–Whitney U tests were used instead. Additionally, since different scales were used to assess levels of depression (BDI-II and GDS) and autonomy (AIADL and IADL), we normalized these scales to standardize their scoring. Normalization was performed by dividing the obtained score by the maximum possible score on the respective scale. This method resulted in a scoring range from 0 to 1 for both depression and autonomy.

We used structural equation modeling to investigate the relationship between psychological, cognitive, functional, and quality of life variables in a sample of 103 participants aged between 55 and 75 years residing in the city of Bergamo.

Unlike the descriptive analyses, the entire sample was included in the model’s creation without division into groups. Age was used as a predictive variable within the model, thus allowing for the analysis of its influence. Normalized data on depression and autonomy were also used in this analysis.

By creating a graphical model, it was possible to explain how observed (depression, autonomy, cognitive reserve, executive functions, and age) and latent variables (quality of life) are interrelated in aging, influencing the path of an individual’s life.

To assess the feasibility of the model, the following indices were considered: the comparative fit index (CFI), the Tucker–Lewis index (TLI), the root mean square error of approximation (RMSEA), and the standardized root mean square residual (SRMR).

A good model is characterized by a CFI and TLI equal to or greater than 0.9 and an RMSEA and SRMR equal to or less than 0.08.

## 3. Results

### 3.1. Descriptives Analysis

To descriptively analyze the performance trends of the participants, we divided the entire sample into two age groups (55–65 and 66–75 years old). The reason for this division is that, according to the literature, there are measurement tools specifically targeted at older individuals, while there are others for the remaining adult population. Consequently, we used different scales to assess depressive levels and daily autonomy skills based on age.

Table 1 presents the mean and median scores obtained by the subjects according to the two age groups.

Regarding the CRI, in the first group, nearly 35% scored in the medium range, 44% scored in the medium–high range, and the remaining portion scored high. In the second group, 35% scored in the medium and medium–high ranges, while the rest scored high (30%). Regarding the FAB15, 96% and 98% of the groups (aged 55–65 years and 66–75 years, respectively) scored above the cutoff of 9.36, meaning that there was good executive functioning across the entire sample.

Regarding depression, 92% of the BDI-II respondents scored between 0 and 13 (no depression), while 88% of the GDS respondents scored between 0 and 5 (no depression). Similarly, the average levels of autonomy were quite high in both groups.

Regarding the WHOQOL-BREF, according to Chen et al. [34], good levels of quality of life are those that exceed 70% of the possible score in its domains (i.e., 15.2). The domain with the best outcomes was physical health (69% of the group aged 55–65 years and 49% of the group aged 66–75 years scored above 15.2). In the psychological domain, only 15% of the group aged 55–65 years and 19% of the group aged 66–75 years achieved scores higher than 15.2. In the relational domain, 19% of the group aged 55–65 years and 39% of the group aged 66–75 years achieved good scores. Additionally, we observed that the older group appeared more satisfied with their social relationships. Finally, in the environmental health domain, approximately 30% of the group aged 55–65 years and 37% of the group aged 66–75 years scored higher than 15.2.

### 3.2. Comparative Analysis

The comparison between age groups was significant for autonomy (U = 916.5, *p* = 0.001). All other variables were not significant (*p* > 0.003). The physical health domain (U 1491.5, *p* = 0.268); psychological domain (t [101] = −0.960, *p* = 0.339); social relationship domain (U = 938.5, *p* = 0.009); environmental domain (t [101] = −1.070, *p* = 0.287); Frontal Assessment Battery 15 (U = 995, *p* = 0.029); Cognitive Reserve Index (t [101] = −0.462, *p* = 0.644); and depression variable (U = 989.5, *p* = 0.025) were not statically significant after the Bonferroni correction. 

### 3.3. Structural Analysis

As shown in Figure 1, our structural model is composed of a single latent variable (quality of life) determined by the four domains of perceived quality of life, depression, the CRIq total score, and autonomy levels, with the FAB15 and age scores as the observed variables. Table 2 shows in detail the results of our final model. 

In a first attempt to answer our hypothesis, we included gender as an observed variable in the model. The model did not reach the minimum fit parameters (CFI = 0.982, TLI = 0.979, RSEA = 0.027, and SRMR = 0.085). 

In our second and final attempt at building our model, we eliminated gender as a variable and found better fit parameters. Consequently, we chose the best fitting model to explain our results.

Indeed, the model was shown to have a good and thus better fit, with a CFI of 0.993 (normal ≥ 0.9), a TLI of 0.989 (normal ≥ 0.9), an RSEA of 0.019 (normal ≤ 0.08), and an SRMR of 0.079 (normal ≤ 0.08). 

The observed variables (four domains) were significantly correlated with the latent variable of quality of life (*p* < 0.001). The domains that had a larger effect on the latent variable were the physical health and the psychological (standardized β = 0.686 and 0.700, respectively).

We observed that the quality of life was negatively (−0.07) correlated (z = −3.807, *p* < 0.001) with depression, indicating that a higher level of quality of life leads to lower depression symptoms. Age was significantly and positively correlated (0.007) with depression (z = 2.788, *p* = 0.005). Furthermore, it is noteworthy that quality of life and age lead to a greater effect on depression (Standardized β = −658 and 0.312, respectively).

We observed that autonomy, total cognitive reserve, and FAB15 scores were not significantly correlated with depression (*p* > 0.05). 

However, the residual correlation showed that there was a positive and significant correlation (4.417) between the total cognitive reserve and quality of life (z = 2.467, *p* = 0.014), demonstrating an association between higher cognitive reserve levels and higher quality of life levels. No significant correlations were identified between autonomy and quality of life. The cognitive reserve index correlated positively (0.203) with autonomy (z = 2.047, *p* = 0.041).

The correlation between FAB15 and CRI was not significant (*p* > 0.05).

## 4. Discussion

This study aimed to investigate the impact of cognitive, functional, and quality of life factors on the emotional state, and the interplay among these, of a late adult and old adult population from a middle-sized city in northern Italy, specifically Bergamo. 

From the results of the comparative analysis conducted on the sample, which was divided into two age groups (55–65 and 66–75), it is possible to observe that the two groups were very similar to each other with respect to most of the variables measured. The only significant difference was found in the autonomy variable.

Since the differences were so limited, we proceeded, in our subsequent analyses, to create a single model for the entire sample.

Our model results, similar to those of Fiske and colleagues [35], show that age and depression have a certain relationship in the aging period. Indeed, in the model, we found a positive and significant correlation between these two factors, establishing that the age its capable of predicting depression levels. Consequently, as age increases, depression symptoms also increase.

Furthermore, based on our results, it emerged that the quality of life can also predict depression in the elderly, but in a negative way. This worrying correlation shows that higher quality of life scores lead to lower depression levels; conversely, a lower quality of life is correlated with a worse emotional state. Previous studies have already demonstrated the key role of depression in aging, which is a very common psychological disease among older people. In addition, it can lead individuals to undergoing cognitive loss [36], and it is often associated with an unhealthy lifestyle [37].

Consistent with Zhou et al. [38], our results suggest that those who have better physical health, psychological well-being, social relationships, and environmental conditions also have a better emotional status.

Although we did not find a significant correlation between cognitive factors and depression in the regression path of our analysis, our model showed that cognitive reserves and autonomy levels are indirectly related to depression. When analyzing the residual correlation among the variables, it is possible to observe that cognitive reserve and quality of life are positively and significantly correlated, which means that people with a higher cognitive reserve had a better quality of life. The same can be said about autonomy and cognitive reserve. In other words, our results suggest that both cognitive reserve and independence in daily living activities could be considered protective factors for older people inasmuch as they improve their quality of life and reduce depression symptoms. 

This is all consistent with the findings of other studies that have sought to understand the influences of various factors on the aging process. Some studies have shown that engaging in hobbies or enjoyable activities during leisure time improves the quality of life of older adults [39]. In addition, the types of occupation of these individuals appears to be highly influential [40]. In our study, both aspects were measured using the CRI questionnaire, which collects information on the type of occupation a person has; the weekly, monthly, and annual frequency of leisure activities; and their level of education, which is also considered a protective factor if it is higher [41].

Regarding the role of autonomy in activities of daily living, while our study highlights the positive relationship between autonomy and the brain’s ability to better cope with change through functional strategies, other studies have pointed out the potential risks associated with compromised independence. One study showed that deficits in daily living activities are correlated with deficits in functional and cognitive status and, similar to our findings, across the four domains of the WHOQOL-BREF. The same study, in perfect agreement with ours, also revealed that impairments in this area, together with factors such as support and social participation, emerged as predictors of the score obtained in the physical health domain [42]. In addition, the possibility of having social resources minimizes depression while improving memory functions [43]. 

Contrarily to what was hypothesized, in the regression path no significant or direct correlations were found between the cognitive, autonomy, and depression factors. 

These unexpected data could derive from the fact that the tools used, especially the FAB15, are simple screening tests that are quick to administer but not complete. For example, this cognitive assessment lacks memory or space–time orientation tasks. Future studies in this area could use more in-depth measures. 

As regards the lack of correlation between the scores of the FAB 15 and the Cognitive Reserve Index, it is worth considering that these measures evaluate two different types of cognitive ability; the first is more related to a fluid and purely cognitive intelligence, while the second questionnaire refers to crystallized intelligence, or competences and abilities acquired from life experiences. The discrepancy between objective measures and self-reports may have affected these results.

In conclusion, this study provides valuable insights into the multidimensional nature of aging and highlights two key necessities. First, it underscores the importance of advancing knowledge in this field by incorporating various facets of being older and experiencing aging, including different perspectives and disciplines. Such an interdisciplinary approach could enrich our understanding and lead to more comprehensive and effective interventions.

Second, the findings emphasize the need for targeted lifespan interventions that enhance the quality of life of old adults and early older adults. These interventions should focus not only on physical health but also on psychosocial components, such as interpersonal relationships, cognitive reserves, and daily autonomy. By fostering a holistic approach to aging, we can better support the well-being and independence of current and future older individuals. Furthermore, having included the late adult population in our model and not just the old, it is possible to note that the protective or risk role of psychological, cognitive, and quality of life factors comes into play even before this stage of life is reached. Consequently, in building the health promotion and frailty prevention interventions that we aspire to, in addition to paying attention to their multicomponent nature, aging should be considered a process that begins well before the age of 65. Moreover, these interventions could be specifically addressed to adult or late adult individuals to raise their chances of achieving future healthy aging, leveraging the factors that already interact with health at an earlier age, such as the cognitive reserve that develops throughout a person’s life span.

Future research should continue to explore the complex interactions among the psychological, cognitive, and contextual factors involved in aging. Longitudinal studies and diverse population samples will be crucial for validating and extending our findings. In addition, there is a need to develop and implement practical strategies that will translate this knowledge into actionable steps to improve the lives of older adults.

In summary, addressing the multifaceted challenges of aging will require concerted efforts by researchers, policymakers, and clinicians. By promoting a deeper understanding of the aging process and implementing interventions that enhance quality of life, we can help ensure that individuals age with dignity, health, and fulfillment.

The present study has some limitations. The sample is mainly composed of women and this may affect our results, which are therefore more sensitive to the female population than to the male population. However, from the perspective of aging, this may be representative of the fact that women appear to be longer-lived than men. Another limitation regarding sampling is related to the fact that the participants in this study were volunteers; they may have been more motivated to respond to our questionnaires due to their perceived good health. Accordingly, the sample may not be representative of the more general late adult and elderly population and the results may not be generalizable.

The insufficient number of participants per group did not allow us to perform multigroup analyses or compare the correlation paths between two models.

The instruments used to measure emotional state, autonomy in daily activities, and perceived quality of life were self-rated scales, the responses of which were purely subjective. Moreover, in this study, different tools were used to assess participants’ emotional state and autonomy in daily life. Although, through the normalization of these scales, intervening variables were eliminated from the scores and the AIADL was found to be positively correlated with the IADL [31], as was the BDI-II with the GDS in a study on older women [44], the use of different scales may imply different cut-offs, especially for monitoring depressive symptoms. Our study contributes to the growing literature on aging, but future research and perspectives may address this limitation and provide more robust findings.

## 5. Conclusions

Our findings contribute to the existing literature on the phenomenon of aging by shedding light on the ways in which factors of diverse origins influence individuals’ aging trajectories. Maintaining a good level of cognitive reserve, daily autonomy, and perceived quality of life appears to be a prerequisite for enjoying healthy longevity, thereby reducing the likelihood of developing depressive pathology.

Furthermore, our study highlights the importance of involving participants who have not yet reached the statistically defined category of old age, emphasizing the need for individuals to think ahead and prepare for their aging journeys by constructing trajectories characterized by optimal health conditions or by changing their habits when they prove maladaptive. The same need applies to clinicians, researchers, and policymakers who, based on these results, could develop targeted interventions to prevent frailty or promote health in aging even before it manifests.

In conclusion, our research underscores the imperative nature of a proactive engagement in healthy aging practices and underscores the importance of early intervention strategies to mitigate the possible and eventual negative outcomes associated with aging. By recognizing and addressing these factors early on, individuals and stakeholders can better prepare for and navigate the challenges of aging, ultimately fostering improved well-being and resilience in older populations. 

## Figures and Tables

**Figure 1 ijerph-21-01117-f001:**
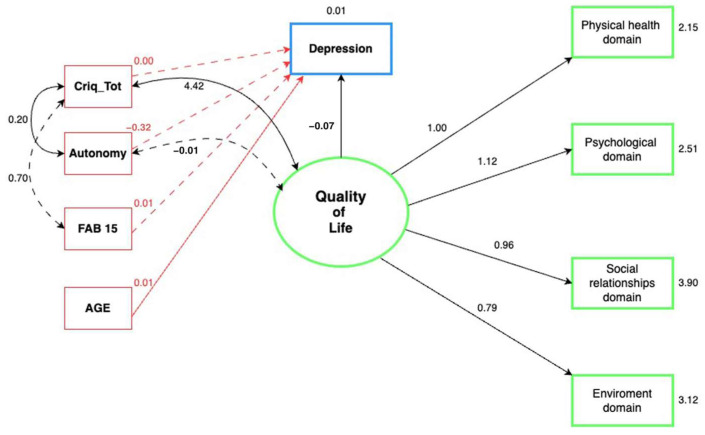
Relationships between quality of life, depression, cognitive reserve, autonomy, executive function, and age in aging. Circles: latent variables; rectangles: observed variables; single arrows: regression and structural model; double, curved arrows: residual correlation; dashed lines: non-significant relationships (*p* > 0.05). Unstandardized beta coefficients were used. Depression was evaluated using the Geriatric Depression Scale and the Black Depression Inventory II. Criq_Tot: total CRI of the Cognitive Reserve Index Questionnaire; FAB15: Frontal Assessment Battery 15. Autonomy was evaluated using the AIADL and IADL scales. The physical health, psychological, social relationships, and environmental domains are from the WHOQOL-BREF questionnaire.

**Table 1 ijerph-21-01117-t001:** Participants’ scores in mean ± SD or median (IQR). The scores in median (IQR) indicate that the data are not normally distributed.

**Variables**	**55–65-Year-Old Age Group** **(n = 52)**	**Males** **(n = 10)**	**Females** **(n = 42)**
CRI-School	110 (14)	117 ± 14.5	111 ± 10.6
CRI-Work	118 ± 12.9	115 ± 9.69	118 ± 13.6
CRI-Leisure Time	115 ± 18.6	109 ± 19.4	116 ± 18.4
CRI	120 ± 11.4	120 ± 11.0	120 ± 11.6
FAB15	12.4 ± 1.44	11.7 ± 1.26	12.6 ± 1.44
AIADL	34 (3)	34.5 (1)	34 (3)
BDI-II	5 (6.75)	3 (10.2)	5 (5.75)
Physical health domain	17 (2)	14.9 ± 1.66	17 (2)
Psychological domain	13.4 ± 2.36	13 ± 2.05	13.5 ± 2.44
Social relationship domain	13 (3)	12.4 ± 2.27	13.9 ± 2.25
Environmental domain	14.4 ± 1.86	14.2 ± 1.40	14.4 ± 1.96
**Variables**	**66–75-Year-Old Age Group** **(n = 51)**	**Males** **(n = 11)**	**Females** **(n = 40)**
CRI-School	113 (17.5)	117 ± 13.2	113 ± 15.2
CRI-Work	113 ± 20.4	113 ± 16.0	113 ± 21.6
CRI-Leisure Time	121 ± 24.1	112 ± 23.1	124 ± 24.0
CRI	121 ± 18.5	118 ± 17.2	122 ± 19.0
FAB15	13.2 (2.19)	12.9 ± 1.21	13.4 (2.29)
IADL	8 (0)	7 (1.5)	8 (0)
GDS	2 (2)	1 (2.5)	2 (3)
Physical health domain	15 (3)	18 (1)	15 (3)
Psychological domain	13.8 ± 2.07	14.5 ± 2.70	13.56 ± 1.85
Social relationship domain	15 (3)	15.7 ± 2.49	15 (3)
Environmental domain	14.8 ± 2.28	15.7 ± 2.87	14.6 ± 2.06

CRI: Cognitive Reserve Index Questionnaire (including CRI-School, CRI-Work, and CRI-Leisure Time); FAB15: Frontal Assessment Battery; AIADL: Advanced Instrumental Activities Daily Living; IADL: Instrumental Activities Daily Living; BDI-II: Beck Depression Inventory; GDS: Geriatric Depression Scale. The physical health, psychological, social relationship, and environmental domains feature in the WHOQOL-BREF Questionnaire.

**Table 2 ijerph-21-01117-t002:** Detailed model results.

**Structural Model**	**Raw ß**	**Stand. Err.**	**z**	** *p* **	**Lower CI (95%)**	**Upper CI (95%)**	**Std. β**	**R^2^**
Depression								0.512
QoL	−0.072	0.019	−3.807	0.000	−0.108	−0.035	−0.658	
Autonomy	−0.319	0.482	−0.663	0.508	−1.264	0.625	−0.137	
FAB15	0.007	0.009	0.749	0.454	−0.011	0.025	0.071	
Cognitive reserve	0.001	0.002	0.537	0.592	−0.002	0.004	0.091	
Age	0.007	0.003	2.788	0.005	0.002	0.013	0.312	
**Measurement Model**	**Raw ß**	**Stand. Err.**	**z**	** *p* **	**Lower CI (95%)**	**Upper CI (95%)**	**Std. β**	**R^2^**
QoL								
Physical health domain	1.000				1.000	1.000	0.686	0.471
Psychological domain	1.123	0.266	4.227	0.000	0.602	1.644	0.700	0.490
Social relationship domain	0.959	0.224	4.281	0.000	0.520	1.398	0.558	0.311
Environmental domain	0.791	0.198	3.990	0.000	0.403	1.180	0.527	0.278

Raw ß: Estimate; Stand. Err: Standardized Error; z: z value; *p*: *p* value; Lower CI (95%): Lower confidence interval; Upper CI (95%): Upper confidence interval; Std. β: Standardized β, used as the effect size; R^2^: R-squared values.

## Data Availability

The original contributions presented in this study are included in the article; further inquiries can be directed to the corresponding author/s.

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
