# Peer review of "A Structural Equation Model for Understanding the Relationship between Cognitive Reserve, Autonomy, Depression and Quality of Life in Aging"

_ijerph, 2024, doi:10.3390/ijerph21091117_

Round 1

Reviewer 1 Report

Comments and Suggestions for Authors

The study explores the interplay among psychological, cognitive, functional, and quality of life factors in late-adult and old adult populations from Bergamo, Italy. I have several questions and suggestions that I hope the authors can address:

1. Section 3.2, "Structural Analysis," seems to blend methods and results. Please ensure that analytical methods are placed within the methodology section.

2. The rationale behind the specified SEM is unclear. Is it informed by existing theoretical frameworks related to population aging, frailty, or cognitive studies? For instance, why is gender not included in the model?

3. It is noteworthy that the authors used different measurements for the age groups 55-65 and 65-75. These groups were combined in the SEM analysis, yet the conclusion emphasizes the “importance of involving participants who have not yet reached the statistically defined category of old age.” This point is quite important for future public health interventions. However, it is unclear how this conclusion was drawn, as there seems to be minimal discussion of this aspect in the SEM results.

4. I am also concerned about the merging of the two age groups, particularly whether the two sets of measurements are comparable for SEM analysis. While it is understandable that the data may have been collected separately initially, adding further discussion and analysis to examine the extent of these differences would be beneficial.

5. What are the potential biases in the sample? The 103 participants were self-selected into study, and there are more females compared to males. How might these factors affect the external validity of the results?

6. The conclusion states that “Our results confirmed that age influences depression.” We should avoid using causal language in correlational studies.

7. Additionally, the research questions seem somewhat vague. Merely stating that the study is "exploring the interplay among various factors" lacks specificity. It would be more informative to outline  the research hypotheses being examined.

Author Response

Comments 1: Section 3.2, "Structural Analysis," seems to blend methods and results. Please ensure that analytical methods are placed within the methodology section.

Response 1: Thank you for pointing this out. We agree with this comment. Therefore, We created an additional paragraph into the Material and Methods section called “ Statistical Analysis” that explains the methods used to all the analyses we performed (Descriptive, Comparative - added after comments below - and SEM). Regarding the SEM everything written in “Structural Analysis” has been moved to the new paragraph. You can find it at page 5 from the line 205 to the line 236 in red. Specifically, the SEM analytical methods are placed from line 222 to 236.

Comments 2: The rationale behind the specified SEM is unclear. Is it informed by existing theoretical frameworks related to population aging, frailty, or cognitive studies? For instance, why is gender not included in the model?

Response 2: Thank you for this question and suggestion. We improved the “Introduction” section of the paper to make theoretical frameworks clearer. You can find it at page 2, from line 73 to line 111 and from 117 to 126 in red.

Regarding including gender into the model, initially we did an attempt including it, as some studies were unclear about the gender impact and we wanted to verify it into the model. However, the fit parameters were CFI= 0.982, TLI= 0.979, RSEA= 0.027 and SRMR= 0.085.

Running another model without the gender, we found better fit parameters. So, we have chosen the better model for our study.

We considered this question as a means to enhance the value of our paper, and decide to reporte this first attempt in the result section. You can find the “Structural section” at page 7, from 286 to 291 lines in red.

Comments 3: It is noteworthy that the authors used different measurements for the age groups 55-65 and 65-75. These groups were combined in the SEM analysis, yet the conclusion emphasizes the “importance of involving participants who have not yet reached the statistically defined category of old age.” This point is quite important for future public health interventions. However, it is unclear how this conclusion was drawn, as there seems to be minimal discussion of this aspect in the SEM results.

Response 3: We agree on that and we explained more in depth this aspect in the Discussion section of the paper, in the attempt to better clarify the implication and the results of having included the late adult (55-65 years) population in the study. You can find this correction at the page 10, from 401 to 410 lines in red.

Comments 4: I am also concerned about the merging of the two age groups, particularly whether the two sets of measurements are comparable for SEM analysis. While it is understandable that the data may have been collected separately initially, adding further discussion and analysis to examine the extent of these differences would be beneficial.

Response 4: Thank you for this suggestion. To address this comment, we added a comparative analysis between the two age groups. You can find the methods used to carry out it into the “2.3 Statistical Analysis” paragraph at page 5 from lines 212 to 221, the Comparative Analysis’ results into the “3.2 Comparative Analysis” paragraph at page 7 from lines 273 to 280 and a brief discussion on these into the “Discussion” section at page 9 from lines 330 to 335. Regarding the two sets of measurement, their scores were normalized in order to eliminate other intervening variables and to be comparable for SEM analysis. Even though we included this aspect as a possible limitation of the study, we also included two studies that found positive correlation between these two sets of measurements.. You can find it at page 11 from lines 433 to 437 in red

Comments 5: What are the potential biases in the sample? The 103 participants were self-selected into study, and there are more females compared to males. How might these factors affect the external validity of the results?

Response 5: We would like to thank the reviewer for this question and reflection. In order to address it, we added a new paragraph in the discussion section relating to the limitations of the research, including the possible biases cited. You can find it at page 11, from line 420 to 439 and specifically the selection bias from the line 420 to the line 427 in red.

Comments 6: The conclusion states that “Our results confirmed that age influences depression.” We should avoid using causal language in correlational studies.

Response 6: We agree with that and changed the wording of the sentence. Now it reads:  “Our results, similar to those of Fiske and colleagues, show that age and depression have a certain relationship in the aging period.” You can find it at page 9, 336-337 lines.

Comments 7: Additionally, the research questions seem somewhat vague. Merely stating that the study is "exploring the interplay among various factors" lacks specificity. It would be more informative to outline the research hypotheses being examined.

Response 7: We agree with this suggestion, and we clarified the research hypotheses at the end of the introduction section. You can find it at page 3, 123-126 lines.

Reviewer 2 Report

Comments and Suggestions for Authors

*Introduction

The relationships among variables for structural equation modeling are not explored with sufficient depth. Existing research findings on the relationships between cognitive reserve, autonomy, depression, and quality of life are not presented in detail. While individual explanations of each variable's impact on aging are provided, there is a lack of discussion on the interactions between these variables. Notably, the expected relationships or hypotheses for variables included in the structural equation model are not clearly presented. The rationale for including these specific variables is not adequately explained, which weakens the foundation for applying structural equation modeling in this study. Addressing these issues would significantly enhance the value of this research.

*Materials and Methods

There is insufficient detail regarding participant recruitment methods and inclusion/exclusion criteria.

Specifically, the rationale for dividing the analysis into 55-65 and 66-75 age groups should be explicitly stated. For instance, individuals aged 65 and above could be categorized as elderly, while those 55 and above might be termed early elderly. However, this study groups ages 66-75 together, necessitating a discussion of this group's representativeness.

The implications of using different assessment tools for the two age groups on result interpretation should be addressed.

Ethical considerations should be mentioned.

The method for handling missing data in the analysis is omitted.

*Results

There is no reporting of effect sizes. It is recommended to specify effect sizes, as practical significance is as crucial as statistical significance.

Confidence intervals are not reported. These are important for assessing the precision of results and should be included.

Information on the model's explanatory power (e.g., R-squared values) is lacking. Additionally, there is insufficient explanation or interpretation of the lack of correlation between FAB15 and CRI.

Notably, there is a lack of analysis on the differences between the two age groups. Presenting a comparison between groups would be beneficial.

*Discussion

The discussion of the study's limitations is inadequate. Particularly, the impact of using different assessment tools for the two age groups should be addressed.

As mentioned earlier, there is insufficient discussion on the relationship between FAB15 and CRI.

Analysis and discussion of differences between the two age groups are lacking.

A more in-depth discussion of the study's limitations and unexpected results, as well as a more cautious approach to the generalizability of the results, is necessary.

Author Response

Comments 1: *Introduction

The relationships among variables for structural equation modeling are not explored with sufficient depth. Existing research findings on the relationships between cognitive reserve, autonomy, depression, and quality of life are not presented in detail. While individual explanations of each variable's impact on aging are provided, there is a lack of discussion on the interactions between these variables. Notably, the expected relationships or hypotheses for variables included in the structural equation model are not clearly presented. The rationale for including these specific variables is not adequately explained, which weakens the foundation for applying structural equation modeling in this study. Addressing these issues would significantly enhance the value of this research.

Response 1: We would like to thank the reviewer for this comment, which gave us the opportunity to strengthen our introduction and consequently increase the value of our paper. We completely agree with the suggestion and for this purpose we have explained in a more profound way the relationships among the variables covered by our model based on existing literature. Furthermore, the research hypotheses were explicitly stated. You can find this part added into the “Introduction” section of the paper at page 2 from line 73 to line 111 and from line 117 to line 126.

Comments 2: *Materials and Methods

There is insufficient detail regarding participant recruitment methods and inclusion/exclusion criteria.

Specifically, the rationale for dividing the analysis into 55-65 and 66-75 age groups should be explicitly stated. For instance, individuals aged 65 and above could be categorized as elderly, while those 55 and above might be termed early elderly. However, this study groups ages 66-75 together, necessitating a discussion of this group's representativeness.

The implications of using different assessment tools for the two age groups on result interpretation should be addressed.

Ethical considerations should be mentioned.

The method for handling missing data in the analysis is omitted.

Response 2: We agree with these comments. We have better outlined the methods for recruiting participants in the "participants" section. You can find the changes made at page 3 from line 141-145. As well, we explicitly state, as suggested, the reason for dividing the two groups from line 135 to line 140 (page 3) and used labels to categorize the two groups in the lines 131-132 in red (page 3).

With regards to the missing data, all of them concerned to WHOQOL-Bref and were handled on the basis of the scoring procedures suggested by the World Health Organization. This has been specified at page 4, lines 149-150. Considering your suggestion as an opportunity to improve the tools and the procedure explanation, we added this scoring procedure at page 5 from line 202 to line 204.

The Ethical considerations were included into the “Material and Methods”, as your suggestion, at page 4 from line 151 to line 153. We agree with you that it is an essential part that should be included in all studies explicitly into the paper.

For what concern the implication of using different assessment tools and the representativeness, we addressed these issues as limitations of the study at page 11 from line 420 to line 439 in red.

Comments 3: *Results

There is no reporting of effect sizes. It is recommended to specify effect sizes, as practical significance is as crucial as statistical significance.

Confidence intervals are not reported. These are important for assessing the precision of results and should be included.

Information on the model's explanatory power (e.g., R-squared values) is lacking. Additionally, there is insufficient explanation or interpretation of the lack of correlation between FAB15 and CRI.

Notably, there is a lack of analysis on the differences between the two age groups. Presenting a comparison between groups would be beneficial.

Response 3: We would like to thank the revisor for this essential recommendation about statistical reporting of the model, that increase for sure the transparency and interpretability of the results. In regard to the effect size, confidence interval and model's explanatory power (e.g., R-squared values), we created a new table with all these parameter. You can find it at page 8, below the model figure. You can also find some brief comments about the magnitude of effects (standardized β) paths into the regression and measurement models at page 8 from line 309 to line 311 and at page 9 lines 315- 316.

Furthermore, as suggested, we included an interpretation of the lack of correlation between FAB15 and CRI. You can find it at page 10 from line 384 to line 389.

As suggested, we proposed a new comparative analysis between the two groups. You can find the methods used to carry out it at page 5, from line 212 to line 221 into the “2.3 Statistical analysis” section, the analysis results at page 7 from line 237 to line 280 into the “3.2 Comparative analysis” section and we also comment on it into the “Discussion” section at page 9 from 330 line to 335 line. Unfortunately, due the insufficient number of participants in each group it was not possible to carry out multigroup analysis to compare two models. . But that could be an interested aspect to be explored in a future study.

Comments 4: *Discussion

The discussion of the study's limitations is inadequate. Particularly, the impact of using different assessment tools for the two age groups should be addressed.

As mentioned earlier, there is insufficient discussion on the relationship between FAB15 and CRI.

Analysis and discussion of differences between the two age groups are lacking.

A more in-depth discussion of the study's limitations and unexpected results, as well as a more cautious approach to the generalizability of the results, is necessary.

Response 4: We agree with these comments. In regard, we added at the end of the discussion a part in which we underlined the limitation of the study, to make the readers aware on this and to make us aware of what we can do better in future studies. You can find it at page 11, from line 420 to line 439 in red.

We discussed more in-depth on the lack of correlation between FAB15 and CRI and about the unexpected results. You can find it at page 10 from line 378 to line 389.

As mentioned in the above comments, we included the analysis and discussion between the two groups comparison.

Regarding the more cautious approach to the generalizability of the results, as well as considering this as a limitation of the study, we have made some changes in the wording of the following sentences to avoid a casual language: page 9, line 341 “it emerged” instead of “we can state”; line 353 “are indirectly related to” instead of “indirectly influence”; line 356 “had” instead of “have” (as the results refer to our sample population, not the general one); and line 358 “could” instead of “should”.

Round 2

Reviewer 2 Report

Comments and Suggestions for Authors

Dear Authors,

I would like to express my sincere appreciation for your diligent efforts in addressing the points raised in the previous peer review. The revised manuscript demonstrates significant improvements across all sections:

Introduction:

The relationships between variables are now elucidated with more comprehensive references to existing research.

The research hypotheses are clearly articulated, providing a robust framework for the study.

Materials and Methods:

The participant recruitment process and inclusion/exclusion criteria are now well-delineated.

The rationale for the age group division is clearly explained.

Ethical considerations have been appropriately addressed.

The method for handling missing data is now explicitly described.

Results:

The inclusion of effect sizes (Standardized β) enhances the interpretation of your findings.

The reporting of confidence intervals adds to the robustness of your results.

The model's explanatory power (R² values) provides valuable context for your analyses.

The comparative analysis between the two age groups offers insightful demographic perspectives.

Discussion:

The added discussion on the study's limitations demonstrates critical self-reflection.

The explanation regarding the lack of correlation between FAB15 and CRI scores provides necessary clarification.

These enhancements have substantially elevated the quality and rigor of your manuscript. Your responsiveness to the peer review process is commendable, and the resultant improvements contribute significantly to the field of aging research.

I extend my gratitude for your dedication to producing high-quality research and for sharing these valuable findings with the scientific community.